# Risk factors of mortality in neonates with neonatal encephalopathy in a tertiary newborn care unit in Zimbabwe over a 12-month period

Hannah Gannon [1,2]*, Gwendoline Chimhini[2,3], Mario Cortina-Borja[1], Tarisai Chiyaka[4], Marcia Mangiza[3], Felicity Fitzgerald[5], Michelle Heys[1,6], Samuel R. Neal[1], Simbarashe Chimhuya[2,3]

1 Population, Policy and Practice Research and Teaching Department, UCL Great Ormond Street Institute of Child Health, University College London, London, United Kingdom, 2 Unit of Child and Adolescent Health, Faculty of Medicine and Health Sciences. Primary Healthcare Sciences, University of Zimbabwe, Harare, Zimbabwe, 3 Sally Mugabe Central Hospital Neonatal Unit, Harare, Zimbabwe, 4 Biomedical Research and Training Institute, Harare, Zimbabwe, 5 Department of Infectious Diseases, Imperial College London, London, United Kingdom, 6 Specialist Children's and Young People's Services, East London NHS Foundation Trust, London, United Kingdom

* Hannah.gannon@nhs.net

**Data Availability Statement:** Researchers interested in accessing the data will first need to

## Abstract

Neonatal encephalopathy (NE) accounts for ~23% of the 2.4 million annual neonatal deaths. Approximately 99% of global neonatal deaths occur in low-resource settings, however, accurate data from these low-resource settings are scarce. We reviewed risk factors of neonatal mortality in neonates admitted with neonatal encephalopathy from a tertiary neonatal unit in Zimbabwe. A retrospective review of risk factors of short-term neonatal encephalopathy mortality was conducted at Sally Mugabe Central Hospital (SMCH) (November 2018 –October 2019). Data were gathered using a tablet-based data capture and quality improvement newborn care application (Neotree). Analyses were performed on data from all admitted neonates with a diagnosis of neonatal encephalopathy, incorporating maternal, intrapartum, and neonatal risk predictors of the primary outcome: mortality. 494/2894 neonates had neonatal encephalopathy on admission and were included. Of these, 94 died giving a neonatal encephalopathy-case fatality rate (CFR) of 190 per 1000 admitted neonates. Caesarean section (odds ratio (OR) 2.95(95% confidence interval (CI) 1.39–6.25), convulsions (OR 7.13 (1.41–36.1)), lethargy (OR 3.13 (1.24–7.91)), Thompson score "11–14" (OR 2.98 (1.08–8.22)) or "15–22" (OR 17.61 (1.74–178.0)) were significantly associated with neonatal death. No maternal risk factors were associated with mortality. Nearly 1 in 5 neonates diagnosed with neonatal encephalopathy died before discharge, similar to other low-resource settings but more than in typical high-resource centres. The Thompson score, a validated, sensitive and specific tool for diagnosing neonates with neonatal encephalopathy was an appropriate predictive clinical scoring system to identify at risk neonates in this setting. On univariable analysis time-period, specifically a period of staff shortages due to industrial action, had a significant impact on neonatal encephalopathy mortality. Emergency caesarean section was

send a request to the Medical Research Council of Zimbabwe; mrcz@mrcz.org.zw.

**Funding:** Funders include the Wellcome Trust: Digital Innovation Award (215742/Z/19/Z to MH) and the Healthcare Infection Society (SRG 201802004 to FF). FF is supported by the Academy of Medical Sciences and the funders of the Starter Grants for Clinical Lecturers scheme. This study and MH and FF are further supported by the National Institute for Health Research, Great Ormond Street Hospital Biomedical Research Centre. The funders had no role in study design, data collection and analysis, or preparation of this report.

**Competing interests:** The authors have declared that no competing interests exist.

associated with increased mortality, suggesting perinatal care is likely to be a key moment for future interventions.

## Introduction

Neonatal mortality contributes to 46% of under-five mortality globally [1]. Of the 2.4 million neonatal deaths that occur annually, the burden of which falls mostly in low-resource settings (LRS), 23% can be attributed to neonatal encephalopathy (NE) [2]. However, data describing risk factors for NE and rates of mortality and morbidity from LRS are scarce [3]. The term NE is used in this setting due to the difficulty in determining cause; the lack of robust evidence of perinatal asphyxia means hypoxic-ischaemic encephalopathy cannot be accurately used. Currently, therapeutic hypothermia is the only treatment option for NE, which is not available in many LRS, where supportive management is the mainstay of treatment. Risk factors for NE have been well described in high-income settings and can be categorised into maternal, antepartum, intrapartum/perinatal and neonatal factors [4–6]. A small number of studies have explored risk factors and outcomes following NE in LRS, most of which are single institution, hospital-based studies [7]. Studies in Nigeria ($n = 150$) [8], Tanzania ($n = 169$) [9] and Kathmandu ($n = 131$) [10] have reviewed specific risk factors. A study in Nigeria using Apgar score at 1 minute to predict short term outcomes reported a 61% survival rate and identified gestational age and birth weight as predictors of survival within their small cohort ($n = 150$) [8]. The Tanzanian study reported a mortality rate of 230 per 1000 and reported associated risk factors as being outborn, low 5 minute Apgar score, depressed clinical status at admission and infection or seizures within the first 24 hours [9]. Ellis *et al.* reported that antenatal risk factors in their cohort in Kathmandu included short maternal stature, high maternal age, primiparity, multiple birth and lack of antenatal care [10]. Further data are needed on the rates of and risk factors for NE-associated mortality and morbidity in LRS in order to identify and mitigate preventable causes of NE and of poor outcome.

In addition to mortality, the burden of morbidity in survivors of NE is likely to be considerable given limited availability of neuroprotective interventions such as therapeutic hypothermia and intermittent availability of supportive medications for NE [6]. Follow-up assessing long-term morbidity and resources for children with disabilities and their families are limited in LRS [7]. Tann *et al.'s* work in Uganda, reviewing early childhood outcomes amongst infants with NE found this group were at high risk of death in early life and neurodevelopmental impairment with significant impact on affected children and their families [11].

There is no gold standard test to diagnosis NE but various factors can be used to support a clinical diagnosis. For example, the presence of a sentinel event, arterial cord blood acidaemia (pH < 7.0), an Apgar score of < 7 at 5 minutes, multi-organ failure, electroencephalogram (EEG) monitoring, supporting brain imaging as well as Magnetic Resonance Imaging (MRI) spectroscopy. While biochemical markers and imaging are available in high-resource settings (HRSs) [12], studies in LRS have demonstrated that clinical examination remains the mainstay of diagnosis [9].

Multiple predictive clinical scoring systems have been developed for identification of neonates with NE but not all are applicable in LRS [13]. The two most widely used scoring systems are the Sarnat scoring system [14] and the Thompson score [15]. The original Sarnat Scoring system, which was initially developed on just 21 newborns, and is still a widely used tool worldwide to identify NE, is dependent on EEG findings. The modified Sarnat score has been

adapted over time and used in many of the therapeutic hypothermia trials [13]. The Thompson score, developed in 1997 was designed to not require specific training or depend on availability of specific diagnostic tools and therefore is suitable in LRSs [15]. It uses clinical signs and symptoms exclusively to diagnose NE. Previous studies have shown the Thompson score has high specificity and sensitivity for adverse outcomes [16].

In this paper we analyse rates of NE in admitted neonates, short-term morbidity and mortality, and maternal, perinatal and postnatal risk factors of poor outcome for a cohort of neonates admitted to a busy referral Neonatal Unit (NNU) in Zimbabwe, over a 12-month period (November 2018 –October 2019). Data were collected during piloting of the Neotree, a tablet-based data capture and newborn care quality improvement system [17].

## Methods

### Study setting

The study was carried out at Sally Mugabe Central Hospital (SMCH), the largest of five tertiary referral hospitals in Zimbabwe. An estimated 12,000 neonates are born annually at SMCH and the 100-cot neonatal unit (NNU) often runs at 140% of its designated bed capacity. It receives high-risk pregnancies from 8 polyclinics and surrounding districts in Zimbabwe's capital city, and is the national neonatal surgical unit. As within many LRS the NNU often functions with limited staff and resources. In the maternity unit there was one functioning theatre during the study period. Medical interns (junior doctors, 1–2 years post-graduation) are responsible for admitting neonates to the neonatal unit and senior support is overstretched. There are limited supplies of anti-seizure medications and other resources for supportive management for neonates admitted with suspected NE. The top three causes of admission to the SMCH NNU (data collected by Neotree) are prematurity, sepsis and NE. Advanced diagnostic tests such as neuro-imaging, biochemical markers and cord blood gas analysis to aid NE diagnosis are not available. The period from November 2018 to January 2019 was characterized by industrial action by doctors in the public hospitals, due to wages and working conditions, which also recurred in September 2019 to February 2020.

The data used for this analysis were prospectively collected through the Neotree project [18], a Wellcome Trust funded (www.neotree.org) digital health platform aiming to improve newborn care and survival in low-resource settings. Neotree offers immediate data capture, clinical decision support and education, and data-driven quality improvement. Although the application is designed for healthcare professionals (HCPs) of all cadres, in SMCH the medical interns complete the admission and discharge forms on a tablet device. The data are captured alongside maternal details. Each new admission form generates a unique Neotree identification number. The pseudonymised data are stored on the tablet and exported to a secure server. The patient identifiable forms are then printed to create the admission and discharge notes for the neonate and the pseudonymised electronic data are uploaded to the electronic database. The unique Neotree identification number is used to match admission and discharge forms to identify outcomes. The pilot version of the application (Neotree-Beta) was found to be highly usable, feasible and acceptable by HCPs in Malawi [17]. The experience and implementation lessons from the pilot study have been reported [18].

Within SMCH there are no facilities for therapeutic hypothermia. Documentation of convulsions is based on HCP observation of active seizures and/or the mother's report. Immediate management on admission involves managing the airway, breathing and circulation. Resuscitation following delivery is either performed by the delivering midwife or receiving intern. Oxygen is given via nasal prongs and neonates are nursed in a normothermic environment in open cots.

Based on neurological symptoms, including tone and activity, and the Thompson score, the assessing doctor grades the severity of NE into three categories with a maximum score of 22.

*Mild NE*: Score: 1–10

*Moderate NE*: Score: 11–14

*Severe NE*: Score:15–22

Initial management depends on severity (see Box 1).

---

### Box 1. Management of NE within Sally Mugabe Central Hospital's neonatal unit.

Mild NE:

- Routine observations
- Enteral feeding by cup or breastfeed
- Monitoring for seizures

Moderate and Severe NE:

- Avoid enteral feeds initially
- Commence on 10% Dextrose infusion,
- Blood glucose monitoring 4–6 hourly.
- Septic screen (blood culture)
- Commence broad spectrum antibiotics

If seizures occur the NNU's protocol is followed with phenobarbitone as first line anticonvulsant.

---

### Design

We carried out a retrospective analysis of patient data over a 12-month period from November 2018 to October 2019 using the Neotree electronic database. We reviewed all matched neonatal admissions and discharges with a diagnosis of NE and the associated maternal, intrapartum and neonatal risk factors.

Inclusion criteria (meets ≥1 of the following):

1. Admission or discharge diagnosis of NE, birth asphyxia or low Apgar score

2. Apgar score at 5 minutes <7

3. Diagnosis of meconium aspiration syndrome AND (associated convulsions OR abnormal neurological signs*)

4. Thompson score >10 at admission (A Thompson score >10 was selected as cut off for moderate disease with reference to the original paper [15].)

*abnormal neurological signs defined as Tone: "high" or "low" or Activity: "coma", "convulsions", "lethargic" or "irritable".

**Exclusion criteria.** We excluded premature neonates under 35 weeks gestation, neonates admitted with gastroschisis or other severe congenital malformations, as per the established definition of NE [6]. We also excluded all unmatched admission and discharge forms as the data available for these neonates were incomplete.

---

Box 2. Risk factors of interest.

**Maternal**:

- *HIV status*: Recorded on admission as either positive, negative or unknown (if not documented or test had not been done at admission)

- *Parity*: Described as nulliparous (0 previous pregnancies) or multiparous (≥1 previous pregnancies)

- *Age*: Maternal age at delivery

- *Number of antenatal clinics attended*: Clinic appointments documented from 0 to >5

*Type of pregnancy*: Singleton, twin, triplet pregnancy

**Perinatal/Intrapartum**:

- *Mode of delivery*: Defined as spontaneous vaginal delivery, elective Caesarean-section, emergency Caesarean-section or Ventouse delivery.

- *Resuscitation required*: Defined as oxygen, stimulation, suctioning, bag-valve mask ventilation (BVM) or Cardio-pulmonary resuscitation (CPR).

- *Inborn vs Outborn*: Defined as neonate delivered in SMCH or outside SMCH facilities

- Duration of labour in hours

- Duration of rupture of membranes in hours

- *Presentation;* Defined as vertex, breech, face, brow or unknown

- *Apgar score at 5 minutes of age*: score 0–9

- *Risk factors for neonatal sepsis*: Defined as PROM>18 hours, maternal fever, offensive liquor, prolonged second stage, born before arrival and prematurity (<37 week)

**Postnatal**:

- Birthweight recorded on admission in grams

- Gestational age

- Sex

**Clinical Neonatal factors**:

- *Tone on admission*: On neurological examination assessment of tone categorised as; "high", "low" or "normal"

- *Thompson score*: Scored 0–22

- *Temperature*: In relation to normothermic, mild, moderate or severe hypothermia as per WHO definitions

---

Primary outcome: was defined as: Death (Neonatal Death = NND) or Discharged (DC). Risk factors of interest (Box 2).

## Data analysis

Data were analysed in the R language and environment for statistical computing, version 3.6.0 (R Core Team, Vienna, Austria) within R Studio version 1.2.1335 (RStudio Team, Boston, United States) [19, 20]. Correlation was assessed using Pearson's $\chi^2$ or Fisher's exact test (Table 1). We fitted univariable and multivariable logistic regression models to analyse the effects of risk factors on neonatal mortality. These models were then used to estimate the effects of each individual variable on neonatal mortality. Variables yielding a *p*-value <0.1 were included in a multivariable model. The results are presented in Table 2 as odds ratios (OR) and adjusted ORs (aOR) alongside with their 95% confidence intervals and *p*-values.

In the final analysis, Model 1 included all predictors with *p* <0.1 in the univariable analysis, Model 2 was additionally adjusted for time-period. The degrees of freedom for regression spline functions were chosen by minimising the Bayesian Information Criterion.

**Missing values.**   Admission and discharge forms were matched using unique Neotree identity numbers. The matching algorithm uses a combination of this unique ID and the date of birth to accurately match. Neonates who did not have a complete set of matched admission and discharge forms were excluded from the analysis. For some of the explanatory variables such as temperature on admission (*n* = 221) and Thompson score (*n* = 143) there was a substantial proportion of missing data. We used multivariate imputation by chained equations as implemented in the R package (mice) [21] to efficiently use all available data. The Neotree dataset captured only the summary Thompson score, manually calculated and inputted by the HCPs, not its individual components. We therefore imputed the summary Thompson score and not its components. We included all potential predictors of neonatal mortality in the imputation model. We generated 30 imputed datasets and estimated pooled odds ratios and their standard errors using Rubin's rules [22]. Collinearity was assessed by Pearson's $\chi^2$ and, using the R library car, generalised variance inflation factors adjusted for the degrees of freedom corresponding to each covariate(GVIFs) [23, 24].

## Ethics statement

The Neotree pilot study received approval from Sally Mugabe Central Hospital Research Ethics Committee (Reference number HCHEC070618/58), University College London Ethics Committee (5019/004), Biomedical Research and Training Institute (AP148/18), the Medical Research Council of Zimbabwe MRCZ/A/2570 and the Electronic Health Records Department of the Zimbabwe Ministry of Health and Child Care. The Neotree follows international and local precedent for collection of pseudonymised data for the purposes of epidemiological surveillance and service evaluation such as the neonatal UK/Australia/New Zealand Badgernet system, or the WHO-led District Health Information Software (DHIS). The need to obtain informed consent was waived as we collected only pseudonymised data routinely documented for clinical care.

## Patient and public involvement statement

Patients and the public were not directly involved in this study. Study findings were shared with implementing partners for broader dissemination.

**Table 1. Demographics and Risk factors associated with neonatal mortality.**

| Variable | Number of neonates ($n = 494$) | Neonatal deaths ($n = 94$) | Proportion of neonatal deaths (%) | $p$-value* |
|---|---|---|---|---|
| Neonatal Demographics | | | | |
| Birthweight (g) | | | | 0.151 |
| 1500-2499g | 41(8%) | 9 | 22 | |
| 2500-3999g | 429 (87%) | 77 | 18 | |
| >/ = 4000g | 19 (4%) | 8 | 42 | |
| Not recorded | 5 (<1%) | 0 | 0 | |
| Gestation (weeks) | | | | 0.196 |
| 35–36$^{+6}$ | 35 (7%) | 8 | 23 | |
| 37–41 $^{+6}$ | 405 (82%) | 74 | 18 | |
| >/ = 42 | 42 (9%) | 12 | 29 | |
| Not recorded | 12 (2%) | 0 | 0 | |
| Sex | | | | 0.512 |
| Male | 314 (64%) | 31 | 10 | |
| Female | 180 (36%) | 63 | 35 | |
| Type of Pregnancy | | | | 0.435 |
| Singleton | 486 (98%) | 92 | 19 | |
| Twin | 8 (1.8%) | 2 | 25 | |
| Triplet | 0 | 0 | 0 | |
| Maternal Demographics | | | | |
| Maternal age (years) | | | | 0.343 |
| 15–19 | 112 (23%) | 19 | 17 | |
| 20–34 | 314 (63%) | 57 | 18 | |
| 35–44 | 68 (14%) | 18 | 26 | |
| Mother HIV status | | | | 0.601 |
| Positive | 48 (10%) | 11 | 23 | |
| Negative | 424 (86%) | 79 | 19 | |
| Not Known | 22 (4%) | 4 | 18 | |
| Maternal parity | | | | 0.581 |
| 0 | 257 (52%) | 46 | 18 | |
| ≥1 | 237 (48%) | 48 | 19 | |
| Number of Antenatal clinic visits | | | | 0.967 |
| ANC: 0 | 30 (6%) | 3 | 10 | |
| 1 Visit | 61 (12%) | 14 | 23 | |
| 2 Visits | 85 (17%) | 14 | 16 | |
| 3 Visits | 114 (23%) | 24 | 21 | |
| 4 Visits | 99 (20%) | 17 | 17 | |
| 5 Visits | 63 (13%) | 12 | 19 | |
| >5 Visits | 42 (9%) | 9 | 21 | |
| Mode of delivery | | | | <0.005 |
| Vaginal | 414 (84%) | 69 | 17 | |
| Caesarean | 80 (16%) | 25 | 31 | |
| Intrapartum Demographics | | | | |
| Inborn vs Outborn | | | | 0.593 |
| Inborn | 365 (74%) | 72 | 20 | |
| Outborn | 129 (26%) | 22 | 17 | |

(*Continued*)

**Table 1.** (Continued)

| Variable | Number of neonates (*n* = 494) | Neonatal deaths (*n* = 94) | Proportion of neonatal deaths (%) | *p*-value* |
|---|---|---|---|---|
| Duration of Labour | 248 (50%) | 49 | 20 | 0.670 |
| ≤ median (9hrs) | | | | |
| > 9 hours | 241 (49%) | 43 | 18 | |
| Not documented | 5 (1%) | 2 | 40 | |
| Presentation | | | | 0.686 |
| Vertex | 437 (89%) | 83 | 19 | |
| Breech | 24 (5%) | 4 | 17 | |
| Face | 10 (2%) | 1 | 10 | |
| Brow | 7 (1%) | 1 | 14 | |
| Unknown | 16 (3%) | 5 | 3 | |
| Duration of rupture of membranes: | | | | 0.444 |
| PROM (>18hrs) | 57 (12%) | 13 | 23 | |
| No PROM | 292 (59%) | 51 | 18 | |
| Not documented | 145 (29%) | 30 | 21 | |
| Apgar Score at 5 minutes | | | | <0.005 |
| 0–3 | 40 (8%) | 29 | 73 | |
| 4–6 | 315 (65%) | 56 | 18 | |
| 7–10 | 130 (27%) | 7 | 5 | |
| Not documented | 9 (2%) | 2 | 22 | |
| Resuscitation | | | | <0.005 |
| None | 32 (7%) | 2 | 6 | |
| Oxygen only | 13 (3%) | 0 | 0 | |
| Stimulation/ Suction | 173 (35%) | 15 | 9 | |
| BVM | 218 (44%) | 50 | 23 | |
| CPR | 36 (7%) | 25 | 69 | |
| Unknown | 22 (5%) | 2 | 9 | |
| Risk factors for neonatal sepsis | | | | 0.741 |
| None | 374 (76%) | 68 | 18 | |
| 1 risk factor | 90 (18%) | 20 | 22 | |
| ≥ 2 risk factors | 29 (6%) | 6 | 21 | |
| Unknown | 1 (<1%) | 0 | 0 | |
| Clinical Neonatal Factors | | | | |
| Tone | | | | <0.005 |
| High | 67 (14%) | 5 | 8 | |
| Low | 144 (29%) | 65 | 45 | |
| Normal | 283 (57%) | 24 | 9 | |
| Neurologic status | | | | <0.005 |
| Alert | 246 (50%) | 16 | 7 | |
| Coma | 17 (3%) | 16 | 94 | |
| Convulsions | 13 (3%) | 5 | 39 | |
| Lethargic | 185 (37%) | 55 | 30 | |
| Irritable | 33 (7%) | 2 | 6 | |
| Thompson Score | | | | <0.005 |
| 1–10 | 329 (67%) | 45 | 14 | |
| 11–14 | 29 (6%) | 16 | 55 | |
| 15–22 | 9 (2%) | 8 | 89 | |
| Not documented | 127 (26%) | 25 | 18 | |

(*Continued*)

**Table 1.** (Continued)

| Variable | Number of neonates (n = 494) | Neonatal deaths (n = 94) | Proportion of neonatal deaths (%) | p-value* |
|---|---|---|---|---|
| Temperature on admission | | | | 0.006 |
| < 36.5°C | 172 (35%) | 51 | 30 | |
| 36.5–37.4°C | 97 (20%) | 13 | 13 | |
| ≥ 37.5°C | 4 (1%) | 0 | 0 | |
| Not documented | 221 (45%) | 30 | 14 | |

Highlighted variables had a p-value <0.1 and were included in the initial regression model

*p-values refer to $\chi^2$ tests

## Results

In total, 3466 neonates were admitted to the NNU during the period under review. Of these, 2894 (84%) had matched admission and discharge records. Our inclusion criteria for diagnosis of NE were met by 758 (meaning 26% (758/2894) of admitted neonates with matched records were admitted with a diagnosis of NE). A further 264 of these 758 neonates were excluded after applying the exclusion criteria leaving 494 neonates for the final analysis (167 neonates were excluded with gestational age < 35 weeks, and 97 neonates were excluded with severe congenital abnormality). See S1 Fig for dataflow chart.

**Table 2.  Univariable ORs (95% confidence intervals) for risk factors and logistic regression models for neonatal mortality on imputed data.**

| Variable | Univariable analysis (n = 494) | | Model 1 (n = 477) | | Model 2 (Adjusted for Time period) (n = 477) | |
|---|---|---|---|---|---|---|
| | OR (95% CI) | p-value | OR (95% CI) | p-value | OR (95% CI) | p-value |
| Mode of Delivery: SVD | 0.20 (0.15–0.25) | <0.001 | Reference | - | Reference | - |
| Mode of Delivery: Assisted | 0.93 (0.21–2.90) | 0.916 | 0.23 (0.03–1.89) | 0.17 | 0.27 (0.03–2.24) | 0.22 |
| *Mode of Delivery: CS* | 2.27 (1.30–3.87) | 0.003 | *2.95 (1.39–6.25)* | *0.005* | *3.14 (1.46–6.78)* | *0.003* |
| Thompson score: 1–10 | 0.16 (0.11–0.21) | <0.001 | Reference | | Reference | |
| *Thompson Score: 11–14* | 7.77 (3.51–17.5) | <0.001 | *2.98 (1.08–8.22)* | *0.04* | *1.95 (0.68–5.58)* | *0.21* |
| *Thompson Score: 15–22* | 50.5 (8.96–947.9) | <0.001 | *17.61 (1.74–178.0)* | *0.02* | *12.34 (1.54–98.7)* | *0.02* |
| Apgar score: 1–3 | 2.6 (1.36–5.51) | 0.006 | Reference | | Reference | |
| Apgar score: 4–6 | 0.08 (0.04–0.17) | <0.001 | 0.25(0.10–0.66) | 0.01 | 0.23 (0.09–0.59) | 0.003 |
| Apgar score: 7–10 | 0.02 (0.01–0.06) | <0.001 | 0.07 (0.02–0.24) | <0.001 | 0.06 (0.02–0.21) | <0.001 |
| Admission Temperature | 0.523 (0.35–0.77) | 0.001 | 0.61 (0.35–1.07) | 0.09 | 0.61 (0.37–0.99) | 0.05 |
| Activity: Alert | 0.06 (0.04–0.11) | <0.001 | Reference | | Reference | |
| *Activity: Convulsions* | 8.98 (2.48–30.30) | <0.001 | *7.13 (1.41–36.1)* | *0.02* | *7.99 (1.63–39.1)* | *0.01* |
| Activity: Irritable | 0.93 (0.14–3.47) | 0.922 | 1.41 (0.27–7.28) | 0.68 | 1.34 (0.25–7.11) | 0.73 |
| *Activity: Lethargic* | 6.08 (3.42–11.37) | <0.001 | *3.13 (1.24–7.91)* | *0.02* | *3.27 (1.31–8.14)* | *0.01* |
| Activity: Coma* | 230 (42.9–4285) | <0.001 | - | - | - | - |
| Tone: High | 0.08 (0.03–0.18) | <0.001 | Reference | | Reference | |
| Tone:Low | 10.2 (4.22–30.48) | <0.001 | 2.95 (0.96–9.11) | 0.06 | 3.02 (0.96–9.54) | 0.06 |
| Tone: Normal | 1.15 (0.45–3.52) | 0.786 | 2.29 (0.71–7.40) | 0.17 | 2.30 (0.70–7.57) | 0.17 |
| Industrial action (First) | 1.08 (0.35–2.72) | 0.890 | - | | 1.47 (0.73–2.97) | 0.28 |
| Industrial action (Second) | 2.14 (1.32–3.43) | <0.001 | - | | 1.53 (0.39–6.02) | 0.55 |

*Coma was not included in the logistic regression models due to the large, independent influence it had on neonatal mortality; 94% dying.

## Neonatal demographics

Singleton pregnancies accounted for 98% of neonates included. A significantly higher proportion were males (64%, $p < 0.05$) though there was no significant difference between sexes related to mortality ($p = 0.5$). Birthweight ranged from 1650g to 4700g, with median 3000g (Interquartile range (IQR) 2800g,3400g), with neonates with birthweights at the extremes of the spectrum at highest risk of death. Fig 1B shows the predicted probabilities of NND as a smooth function of birthweight using a natural cubic spline with 2 degrees of freedom.

Gestational age ranged from 35 weeks (defined in the exclusion criteria) to 43 weeks and 82% of the neonates were born at term gestation (37–41$^{+6}$ weeks). Determination of gestational age was based on the Last Menstrual Period (LMP) in 86% of infants. Abdominal Ultrasound (USS) dating was used in 8% and the remaining 6% was based on fundal height.

## Maternal demographics

Maternal age ranged from 15 to 43 years, median 24 years old (IQR 20, 29), and 23% were adolescent mothers (15 to 19 years old). The majority (94%) received at least one antenatal assessment prior to delivery. The average number of antenatal care reviews was 3, with 64% having three or more assessments. There were 10% of mothers diagnosed with pregnancy-induced hypertension (PIH), 1% were anaemic, and 10% were HIV positive, 1 mother had gestational diabetes.

Of the proportion of deliveries that were Caesarean sections 90% were emergencies.

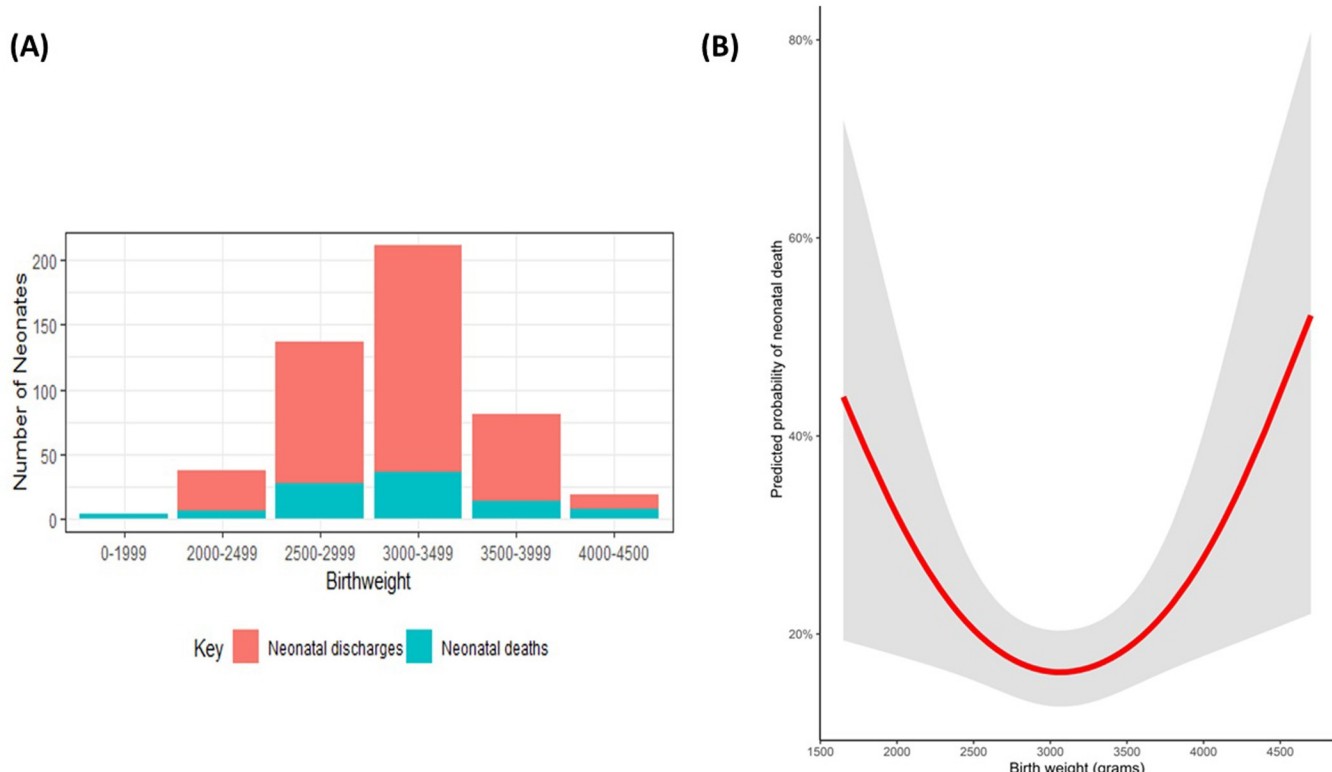

**Fig 1.** (A) Range of Birthweights of neonates with NE over the 12-month period; (B) Predicted probabilities of NND as a smooth function of birthweight (red) with 95% prediction limits (shaded area).

### Intrapartum demographics

Duration of labour was recorded in 91% of the deliveries. Duration ranged between 1 hour to a maximum of 52 hours, median 9 hours (IQR 6–14). Prolonged second stage of labour (defined as $\geq$ 30 minutes) was present in 7.3%. The Apgar score at 5 minutes was recorded in 98% of entries; 72% of scores were less than 7 and 8% (40/485) less than 4, 69% of the neonates receiving CPR died.

In 76% of infants no risk factors for sepsis were present. Only 6% of the neonates had 2 or more risk factors for sepsis, and of this number 21% died.

Blood sugar measurement on admission was recorded in only 3% of neonates due to unavailability of equipment to do bedside glucose monitoring.

### Clinical neonatal factors

Tone and activity on admission were recorded in all 494 neonates, see Table 1 for associated outcomes. The Thompson score was documented in 74% of neonates. Thompson score of >10 was documented in 10%, of whom 64% died. The majority of neonates were inborn (74%), with delivery location not significantly associated with mortality. The Thompson score was not recorded in 28% of the outborn neonates and 24% of the inborn neonates.

Documentation of body temperature on admission significantly improved over the course of the 12-month period, this was after implementation of hypothermia/thermoregulation teaching and the provision of more thermometers for the medical interns on duty in April 2019. Documentation increased from 11% to 93% between November 2018 and June 2019, overall temperature was recorded in 273/494 (55%) admissions.

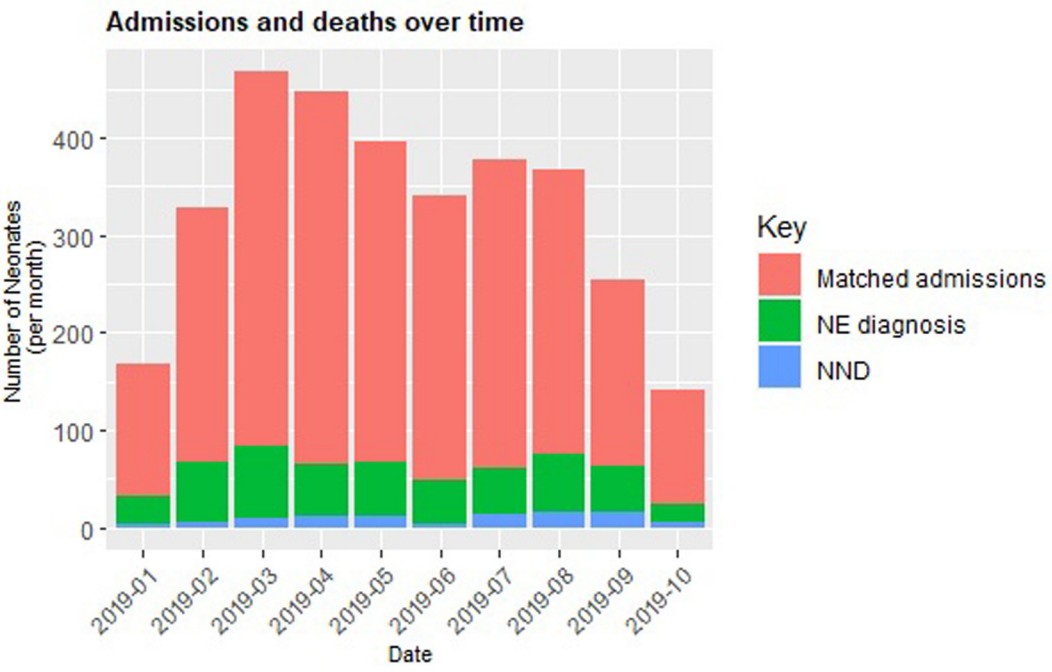

**Fig 2. NE admissions and deaths over a 12-month period.**

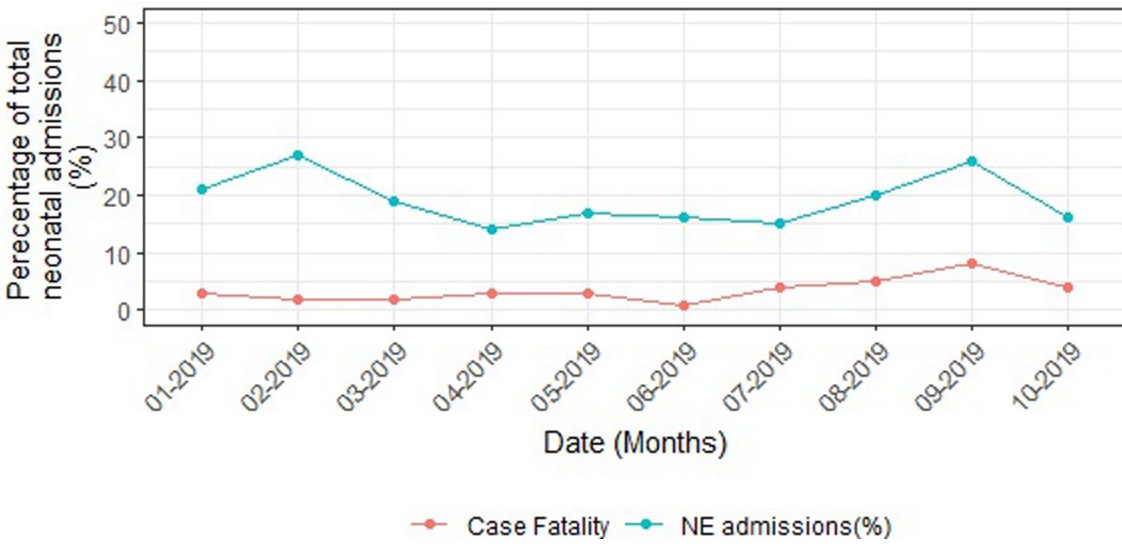

**Fig 3. NE mortality trend over a 12-months period; calendar adjusted.**

### Neonatal outcomes in hospital

There were 94 neonatal deaths (NND) of the 494 neonates during the 12-month period, giving a case fatality rate of 190 per 1000 neonates. The trend of monthly mortality is shown in Figs 2 and 3.

Table 1 describes the relationship between the predictor variables and the primary outcome of neonatal death. Highlighted variables had a significance of $p<0.1$ and were included in the initial regression model. Table 2 shows the unadjusted odds ratios (95% confidence intervals) for the variables significantly associated with neonatal mortality, and the odds ratios from the final logistic regression models fitted using multiple imputation. In the logistic regression models "coma" was removed due to the large independent influence it had on neonatal mortality;94% of the babies presenting with "coma" on examination died, therefore we analysed data from $n = 477$ neonates. On univariable analysis the second period of industrial action was significantly associated with neonatal mortality for neonates with NE (OR 2.14(95% CI 1.32–3.43)). In the final logistic regression model mode of delivery as Caesarean Section (OR 2.95 (95% CI 1.39–6.25), having convulsions (OR 7.13 (1.41–36.1)) or being lethargic (OR 3.13 (1.24–7.91)), Thompson score "11–14" (OR 2.98 (1.08–8.22)) or "15–22" (OR 17.61 (1.74–178.0)) were significantly associated with neonatal death, time period was no longer statistically significant. The majority of hypothesised risk factors were not statistically significant on multivariable analysis. Regarding collinearity in the model, firstly chi-squared tests showed, as suspected, significant associations between categorised Thompson Score and tone and activity. However, generalised variance inflation factor values indicated that these associations did not have an influential impact on the model's power.

### Discussion

In this retrospective review of risk of mortality of neonates with NE admitted to a large, low-resource hospital in Zimbabwe, we found a case-fatality rate of 190 per 1000. This is in the context of limited interventions such as lack of blood gas monitoring, EEG monitoring or therapeutic hypothermia. This is lower than other studies exploring outcomes following NE in LRS, most of which have been smaller single institution, hospital-based studies. A Ugandan study

reported a case fatality rate of 260 per 1000 [25] and Cavallin et al reported a mortality rate of 230 per 1000 in Tanzania [9]. In comparison to high-income settings, recent studies have estimated a case fatality rate of 15%; comprising 27 regional NNUs in the United States [26].

Potential risk factors identified as predictors of mortality within our cohort included neurological activity assessed as coma, lethargic or convulsions, mode of delivery as emergency Caesarean-section and a Thompson score of 11–22. Assessment of neurological activity as coma on admission was significantly associated with neonatal death, the impact on the regression models required its removal, as stated above. This would suggest neonates with coma on admission are at significant increased risk of death, this knowledge can help guide and target practice within the unit.

Maternal factors such as age, number of ANC visits or parity were not significantly associated with worse neonatal outcome in this cohort. Elsewhere, due to the longer duration of first labour, nulliparous women are at greater risk of infection, fetal distress, invasive interventions, operative deliveries, and associated maternal and fetal birth trauma [27, 28]. It may be that the frequency of ANC visits has been protective in this cohort, as was found in Kathmandu [10]. In the future it would be useful to review the partograms of these neonates to assess the relationship with neonatal outcomes.

Mortality was however associated with emergency Caesarean section as mode of delivery with an almost 3-fold increased risk of neonatal death following Caesarean section. This could be explained as these neonates were likely more compromised than those delivered by vaginal delivery, requiring the need for surgical intervention. Most Caesarean sections (90%) were emergencies due to fetal distress. Significant time may lapse between decision to perform a Caesarean section and the actual time when it is performed due to various factors including shortage of operating facilities at the hospital. These findings confirm findings from previous studies [4, 29]. However, unlike the findings of Ezenwa et al's study in Nigeria, outborn neonates did not appear to be at higher risk of neonatal death compared to their inborn counterparts [30]. This could be explained by the differences in referral pathways in Nigeria and Zimbabwe to their tertiary neonatal units, but further research into this field is required.

We found that the Thompson score, although incompletely recorded, was a strong predictor of mortality. Neonates assessed to have moderate NE ($n$ = 29, Thompson score 11–14) or severe NE ($n$ = 9, Thompson score 15–22) were at a significantly increased risk of death. We used a cut-off of 10 for mild NE based on the inception paper by Thompson et al. [15]. However, Horn et al demonstrated that an early Thompson score, recorded before 6 hours of age ≥7 had the best sensitivity and specificity (100% and 66.7%, respectively) to predict an abnormal 6 hour amplified electroencephalograph representing moderate to severe encephalopathy [31]. Our study population had their Thompson scores assessed at admission, although exact timing of assessment was not recorded. It would be useful in further studies to assess the timing of the Thompson score and the impact on outcome. We also did not analyze the peak Thompson score, i.e. the maximum score assessed during the whole admission which is also useful in prediction of neurodevelopmental outcome [15]. In subsequent studies we will review the timings of the Thompson score, the peak Thompson score and consider assessment of scores >7 to assess the impact on the prediction of short-term morbidity and mortality and long-term neurodevelopmental outcomes.

Another interesting variable of the dataset was the time frame selected itself, characterized by two periods where there were staff shortages as a result of industrial action, and the impact on NE. Our findings indicate that NE mortality was significantly associated on univariable analysis with the second period of industrial action, although the full length of the second period of industrial action was not captured within this dataset. This could be explained by the impact of reduced staffing to manage complex deliveries had a direct impact on neonatal outcome, as described by Bikwa et al during the first wave of the COVID-19 pandemic [32]. However, this finding was attenuated to the null after adjusting for other potential risk factors.

Further review of the ongoing data collection would allow re-assessment of the full impact of unavailability of staff on NE and the risk factors associated with mortality. The assumption would also be that the data would be increasingly complete relieving the need for imputation. There is a gap in the literature describing the impact of extrinsic challenges, such as pandemics or industrial action, to NE morbidity and mortality outcomes and further review is needed to develop mitigating strategies.

We know that early effective resuscitation in the delivery room may reduce birth related mortality in LRS [12, 33]. It has been well documented the significant impact of delaying resuscitation on mortality rates, 69% of the neonates in this study requiring CPR died. Following the introduction of the Helping Babies Breathe programme in Tanzania they reported a 47% reduction in 24-hour mortality [34]. Fetal heart rate monitoring is one method of potentially detecting these high-risk neonates. In LRS this is frequently performed by intermittent auscultation of the fetal heart rate by stethoscope. Recent developments of low-cost foetal heart rate monitors could help. A recent study however found that despite detection of these high-risk neonates time to Caesarean section was the same across both groups [35], suggesting resource availability, i.e. theatre and practitioner availability, also plays a large role in mortality outcomes in these settings. Future studies into the timings of both resuscitation and from decision made to performance of Caesarean section within this cohort would be essential. Interventions to target safer delivery outcomes with a focus on accessible quality neonatal resuscitation and HCP development could then be developed. Data capture, analysis and ongoing informed quality improvement are essential for improving health services [36]. The data used in this study, prospectively collected by the Neotree, is a rich resource of information which can be used by the local teams to inform ongoing quality improvement projects.

The data and results from this paper will be presented to the management board at SMCH and Ministry of Health and Child Care to leverage the need for the provision of essential drugs and resources for the supportive management of these NE neonates. Following the findings of this study we are in discussion with the Obstetric team in how best to implement quality improvement projects with the aim of improving the intrapartum care of these at risk neonates.

## Limitations

The dataset is dependent on all neonates admitted and discharged from the unit being captured on the Neotree database, which may not have been the case during initial implementation. However, the completeness of the dataset significantly improved over the first 3–4 months of the Neotree study once the application had been fully imbedded in everyday practice, this was confirmed when monthly admission and discharge numbers were compared directly to the previous "gold standard" of the ward book, located in the NNU. Neonates who die prior to admission immediately post-partum will be missed in this data set, which may prove to be an important subset of neonates with NE and may explain the lower mortality rate described here. The analysis presented here demonstrates the need to collect stillbirth and labour ward outcomes in addition to outcomes after admission to NNU and this is a focus for the next phase of Neotree implementation, commenced during the COVID-19 pandemic from June 2020. Additionally, a proportion of neonates were excluded due to using matched admission and discharges only. There were 177 neonates who had only a discharge form completed with a diagnosis of NE and were excluded from the analysis as it provided an incomplete dataset. The demographics of these neonates did not significantly vary from the neonates included in the analysis. We accounted for missing variables with multiple imputation where possible. Long term neurodevelopmental outcomes were not assessed in our study.

### Strengths

This is one of few in-depth data sets collected from a neonatal unit in an LRS.

### Conclusion

Our findings indicate that 1 in 5 neonates admitted to a tertiary LRS NNU with NE died before discharge. We found clinical features and emergency Caesarean section delivery most predictive of poor outcome, which emphasizes the importance of improving perinatal as well as postnatal care in LRS in order to reduce global mortality from neonatal encephalopathy. Improvements in perinatal care could be achieved through data capture and quality improvement systems, such as the Neotree, that focus on adherence to best practice. These data also suggest postnatal supportive care should be optimized for neonates with NE who have a high Thompson score, convulsions and who are born by Caesarean section. It is crucial to expand our understanding of the context and causes of NE in LRS if we are to meet the Sustainable Development Goals for global neonatal mortality; to less than 12 per 1,000 live births by 2030.

### Supporting information

**S1 Fig. Data flowchart.**
(DOCX)

**S1 Checklist. STROBE statement—Checklist of items that should be included in reports of *cohort studies*.**
(DOCX)

### Acknowledgments

We are very grateful to the families at Sally Mugabe Central Hospital Neonatal unit and the staff members for their enthusiasm and commitment to the Neotree project, without which this work would not be possible.

### Author Contributions

**Conceptualization:** Hannah Gannon, Gwendoline Chimhini, Michelle Heys, Simbarashe Chimhuya.

**Data curation:** Hannah Gannon, Tarisai Chiyaka, Simbarashe Chimhuya.

**Formal analysis:** Hannah Gannon, Mario Cortina-Borja, Samuel R. Neal, Simbarashe Chimhuya.

**Funding acquisition:** Felicity Fitzgerald, Simbarashe Chimhuya.

**Investigation:** Hannah Gannon.

**Methodology:** Hannah Gannon, Mario Cortina-Borja, Marcia Mangiza, Samuel R. Neal, Simbarashe Chimhuya.

**Supervision:** Felicity Fitzgerald, Michelle Heys.

**Visualization:** Hannah Gannon, Simbarashe Chimhuya.

**Writing – original draft:** Hannah Gannon, Gwendoline Chimhini, Marcia Mangiza, Felicity Fitzgerald, Michelle Heys, Simbarashe Chimhuya.

**Writing – review & editing:** Hannah Gannon, Gwendoline Chimhini, Mario Cortina-Borja, Tarisai Chiyaka, Marcia Mangiza, Felicity Fitzgerald, Michelle Heys, Samuel R. Neal, Simbarashe Chimhuya.

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
