## [Decision Letter · Decision Letter 0]

18 Aug 2022

PGPH-D-22-00772

Risks of neonatal mortality in neonates with neonatal encephalopathy in a tertiary newborn care unit in Zimbabwe over a 12-month period

Dear Dr. Gannon,

Thank you for submitting your manuscript to PLOS Global Public Health. After careful consideration, we feel that it has merit but does not fully meet PLOS Global Public Health’s publication criteria as it currently stands. Therefore, we invite you to submit a revised version of the manuscript that addresses the points raised during the review process.

The authors have a communicated a well done study to identify the risk factors of mortality in neonates with encephalopathy.

Overall the reviews have been positive, but there are some clarifications and corrections required before this manuscript can be considered further. I have reviewed the manuscript myself and have some additional doubts which I list below.

Considering the fact that most suggestions are minor, I'm sure the authors will be able to address them in their next version of the submission.

We look forward to receiving your revised manuscript.

Kind regards,

Ramachandran Thiruvengadam, M.D.,

Academic Editor

Journal Requirements:

1. Please ensure that the funders and grant numbers match between the Financial Disclosure field and the Funding Information tab in your submission form. Note that the funders must be provided in the same order in both places as well.

2. Please update your online Competing Interests statement. If you have no competing interests to declare, please state: “The authors have declared that no competing interests exist.”

3. Please provide separate figure files in .tif or .eps format only and remove any figures embedded in your manuscript file. Please also ensure that all files are under our size limit of 10MB.

4. We have noticed that you have uploaded Supporting Information files, but you have not included a list of legends. Please add a full list of legends for your Supporting Information files after the references list.

Additional Editor Comments (if provided):

1. In the conclusion section of the abstract, the authors state, “The Thompson score, a validated, sensitive and specific tool for diagnosing neonates with NE was a good predictor of worse outcomes in this setting.”. To be able to call a characteristic as a good predictor one ought to evaluate its discrimination (AUROC, sensitivity, specificity etc), calibration and clinical usefulness analysis which were not performed in this article. It would be more appropriate to call Thompson score as an independent risk factor than a good predictor at this stage.

2. The rationale behind exclusion criteria aren’t explained, particularly for the preterm < 35 weeks of gestation.

3. In table-1, I see many continuous variables being categorized and univariate analysis has been performed. Why they were done so and not considered as continuous variables?

4. Please add STROBE checklist in the supplementary file.

Reviewers' comments:

Reviewer's Responses to Questions

**Comments to the Author**

1. Does this manuscript meet PLOS Global Public Health’s publication criteria? Is the manuscript technically sound, and do the data support the conclusions? The manuscript must describe methodologically and ethically rigorous research with conclusions that are appropriately drawn based on the data presented.

Reviewer #1: Partly

Reviewer #2: Yes

Reviewer #3: Yes

2. Has the statistical analysis been performed appropriately and rigorously?

Reviewer #1: I don't know

Reviewer #2: Yes

Reviewer #3: Yes

3. Have the authors made all data underlying the findings in their manuscript fully available (please refer to the Data Availability Statement at the start of the manuscript PDF file)?

Reviewer #1: No

Reviewer #2: Yes

Reviewer #3: No

4. Is the manuscript presented in an intelligible fashion and written in standard English?

Reviewer #1: Yes

Reviewer #2: Yes

Reviewer #3: Yes

5. Review Comments to the Author

Reviewer #1: PGPH-D-22-00772

Quality improvement in neonatal services is vital in low resource settings especially in sub-Saharan Africa. The paper presents a detailed review of risk factors for one of the major causes of neonatal deaths.

Specific comments:

Inclusion criteria “Diagnosis of Meconium aspiration syndrome AND associated convulsions” should be clarified. Does it mean infants with MAS are only included if they are observed to have convulsions? What if these infants have neurological signs and high Thompson score but were not observed to have seizures?

Risk factors of interest: Gender?? Do the authors mean sex? If not how was the assignment of gender determined at birth?

Table 1: gestational age < 37 weeks does not accurately describe the study population. It should be revised to 35 – 37 weeks.

Table 1: Risk factors for neonatal sepsis: What does “1 x risk factor” mean? Same style of reporting should be used in each section of the Table.

Conclusions: The authors have not presented data to inform the conclusion that “Improvements in perinatal care could be achieved through data capture and quality improvement systems, such as the Neotree, that focus on adherence to best practice.” The role of Neotree in this study is not explicit.

Reviewer #2: The objective of this study was to determine the case fatality and associated risk factors of NE from a single center in Zimbabwe. These are important data to support the burden of NE and it’s risk factors from an LMIC. 

Findings support conclusions and the manuscript is well written. I only have few minor comments-

Title: Substitute risk with risk factors

-Line 94- Please specify what is HRSs

-Line 122- You mention lack of services. Would help to leverage some of this data to make a case a for provision of these services and highlight this in your discussion.

-Regarding MICE- Did you include a summary measure of the entire Thompson scale, or also examined the missingness at both the item and summary score of Thompson? In making this decision, it may be helpful to examine the missingness at both the item and summary level. If there is very little item missingness within each scale (i.e. if some items for a scale are observed then they are all observed) and analysts will use only the summary scales, then imputing the summary scales may make sense. Graham (2009) suggests that creating and imputing a scale score is appropriate when at least half of the items are observed, the items have high coefficient alphas, and all of the item‐total correlations are similar. In contrast, if these conditions are not met it may make sense to impute the items themselves and then construct the summary scales using the observed and imputed data. (https://www.ncbi.nlm.nih.gov/pmc/articles/PMC3074241/)

-Line 310-Could the lower CFR be due to your explicit exclusion criteria?

-Alluding the authors to this imp paper to consider, if relevant https://www.ncbi.nlm.nih.gov/pmc/articles/PMC8075215/

Reviewer #3: The authors report risk factors for neonatal mortality in encephalopathic neonates at a single unit in Zimbabwe. Overall, the paper is well-written and provides relevant information from a low-resource setting. I have a few comments and questions:

1. The authors mention in the abstract and elsewhere (page 2, page 3) that most deaths due to neonatal encephalopathy occur in sub-Saharan Africa. This is quite far from the truth, although the burden in that part of the world is high. Please refer to the article by Lawn et al, Lancet 2005 more carefully (reference 2 in your manuscript).

2. The term "neonatal encephalopathy" is used throughout instead of hypoxic-ischemic encephalopathy (HIE), although the authors go on to describe therapeutic hypothermia as the intervention for this condition. Neonatal encephalopathy could be due to metabolic disorders, stroke and other conditions. If that term is used because of difficulty in diagnosing HIE specifically given the local context, please state that explicitly.

3. The authors have described the Sarnat score as being EEG-dependent (page 4, line 100). Most trials (and many observational studies) involving infants with HIE in recent years have used a modified Sarnat score. It does not require an EEG.

4. The meaning of the subheading "Inclusion criteria (to be met by 1 or more)" on page 7 is not clear.

5. Among the variables in the logistic regression model were activity (including lethargy/seizures), tone and Thompson score. It is worth noting that Thompson score itself incorporates these other variables. Did the authors note any multicollinearity?

6. PLOS authors have the option to publish the peer review history of their article (what does this mean?). If published, this will include your full peer review and any attached files.

**Do you want your identity to be public for this peer review?** For information about this choice, including consent withdrawal, please see our Privacy Policy.

Reviewer #1: No

Reviewer #2: **Yes: **Fyezah Jehan

Reviewer #3: **Yes: **Nishad Plakkal

---

## [Decision Letter · Decision Letter 1]

4 Nov 2022

Risk factors of mortality in neonates with neonatal encephalopathy in a tertiary newborn care unit in Zimbabwe over a 12-month period

PGPH-D-22-00772R1

Dear Dr Gannon,

We are pleased to inform you that your manuscript 'Risk factors of mortality in neonates with neonatal encephalopathy in a tertiary newborn care unit in Zimbabwe over a 12-month period' has been provisionally accepted for publication in PLOS Global Public Health.

Best regards,

Ramachandran Thiruvengadam, M.D.,

Academic Editor

Reviewer Comments (if any, and for reference):

Reviewer's Responses to Questions

**Comments to the Author**

1. If the authors have adequately addressed your comments raised in a previous round of review and you feel that this manuscript is now acceptable for publication, you may indicate that here to bypass the “Comments to the Author” section, enter your conflict of interest statement in the “Confidential to Editor” section, and submit your "Accept" recommendation.

Reviewer #1: All comments have been addressed

Reviewer #3: All comments have been addressed

2. Does this manuscript meet PLOS Global Public Health’s publication criteria? Is the manuscript technically sound, and do the data support the conclusions? The manuscript must describe methodologically and ethically rigorous research with conclusions that are appropriately drawn based on the data presented.

Reviewer #1: Yes

Reviewer #3: Yes

3. Has the statistical analysis been performed appropriately and rigorously?

Reviewer #1: I don't know

Reviewer #3: Yes

4. Have the authors made all data underlying the findings in their manuscript fully available (please refer to the Data Availability Statement at the start of the manuscript PDF file)?

Reviewer #1: (No Response)

Reviewer #3: No

5. Is the manuscript presented in an intelligible fashion and written in standard English?

Reviewer #1: Yes

Reviewer #3: Yes

6. Review Comments to the Author

Reviewer #1: None

Reviewer #3: The authors have not shared the data from Neotree, but have explained this in the data policy statement.

7. PLOS authors have the option to publish the peer review history of their article (what does this mean?). If published, this will include your full peer review and any attached files.

**Do you want your identity to be public for this peer review?** For information about this choice, including consent withdrawal, please see our Privacy Policy.

Reviewer #1: No

Reviewer #3: **Yes: **Nishad Plakkal
